# Molecular basis for inhibiting human glucose transporters by exofacial inhibitors

Nan Wang [1], Shuo Zhang[1], Yafei Yuan [1], Hanwen Xu[1], Elisabeth Defossa[2], Hans Matter[2], Melissa Besenius[2], Volker Derdau [2], Matthias Dreyer [2], Nis Halland [2], Kaihui Hu He[3], Stefan Petry[2], Michael Podeschwa[2], Norbert Tennagels[4], Xin Jiang[1,5 ✉] & Nieng Yan [1,6 ✉]

Human glucose transporters (GLUTs) are responsible for cellular uptake of hexoses. Elevated expression of GLUTs, particularly GLUT1 and GLUT3, is required to fuel the hyperproliferation of cancer cells, making GLUT inhibitors potential anticancer therapeutics. Meanwhile, GLUT inhibitor-conjugated insulin is being explored to mitigate the hypoglycemia side effect of insulin therapy in type 1 diabetes. Reasoning that exofacial inhibitors of GLUT1/3 may be favored for therapeutic applications, we report here the engineering of a GLUT3 variant, designated GLUT3exo, that can be probed for screening and validating exofacial inhibitors. We identify an exofacial GLUT3 inhibitor SA47 and elucidate its mode of action by a 2.3 Å resolution crystal structure of SA47-bound GLUT3. Our studies serve as a framework for the discovery of GLUTs exofacial inhibitors for therapeutic development.

[1] State Key Laboratory of Membrane Biology, Beijing Advanced Innovation Center for Structural Biology, Tsinghua-Peking Joint Center for Life Sciences, School of Life Sciences, Tsinghua University, Beijing 100084, China. [2] Sanofi-Aventis Deutschland GmbH, R&D, Integrated Drug Discovery, Industriepark Höchst, 65926 Frankfurt am Main, Germany. [3] Sanofi-Aventis Deutschland GmbH, R&D, CMC Synthetics Early Development Analytics, Industriepark Höchst, 65926 Frankfurt am Main, Germany. [4] Bayer AG, R&D, Pharmaceuticals, TA Endocrinology, Metabolism & Reproductive Health, DIU Exploratory Pathobiology, 42096 Wuppertal, Germany. [5] School of Biotechnology and Biomolecular Sciences, The University of New South Wales, Sydney, NSW 2052, Australia. [6] Present address: Department of Molecular Biology, Princeton University, Princeton, NJ 08544, USA. ✉email: xin.jiang@unsw.edu.au; nyan@princeton.edu

The major facilitator superfamily (MFS) glucose transporters (GLUTs), encoded by the *SLC2* genes, mediate the transmembrane movement of various hexoses and derivatives in different tissues[1]. Among the fourteen members in the SLC2 family, GLUT1-4 play a pivotal role in metabolic homeostasis for their housekeeping functions in cellular glucose uptake[2,3]. Dysfunction of GLUTs is associated with a number of disorders, such as the GLUT1 deficiency syndrome and the Fanconi–Bickel syndrome[4–6]. Reduced localization of GLUT4 to the plasma membrane, mostly caused by abnormal insulin signaling, is directly related to the type 2 diabetes mellitus[6,7]. Elevated expression of GLUT1 and GLUT3 has been observed in various cancer types[8,9].

Glycolytic cancer cells mainly rely on aerobic glycolysis for energy supply, known as the Warburg effect. Glucose uptake represents a rate-limiting step for cell hyperproliferation[10]. Upregulated expression of GLUTs meets the increased demand for glucose in cancer cells, thereby opening the opportunities for inhibiting GLUTs as a potential strategy for the treatment of cancer[11–14]. On the other hand, insulin conjugated to GLUT inhibitors has been employed for drug depots, half-life extension, and drug release, a strategy to mitigate the insulin-induced hypoglycemia in type 1 diabetic mouse model[15].

High-resolution structures of a target protein in complex with lead compounds can be invaluable to drug discovery. As a proof of principle, we recently succeeded in deriving an antimalarial compound with improved potency and lowered cytotoxicity based on the substrate and inhibitor-bound structures of the hexose transporter PfHT1 from *Plasmodium falciparum*[16]. Inspired by this achievement, we aim to exploit the structures of human GLUT1 and GLUT3 to contribute to drug discovery.

Crystal structures of human GLUT1 and GLUT3 have collectively revealed the molecular detail for substrate binding and transport mechanism[17,18]. The MFS core fold of GLUTs consists of two repeats, namely the N and the C domains, each containing six transmembrane segments, TMs 1–6 in the N domain and TMs 7–12 in the C domain[19,20]. The two domains, which exhibit a 2-fold pseudosymmetry around an axis that is perpendicular to the membrane plane, enclose a central cavity for substrate accommodation[21]. Exposure of the substrate-binding site to either side of the membrane is achieved through rocker-switch rotation of the two domains as well as local conformational changes exemplified by the bending of TM7[22].

The structures suggest that inhibitors can be rationally designed to either impede substrate binding or interfere with the conformational changes required for achieving the alternating access of the transporter[17,18,23,24]. The ligands that specifically bind to the outward- or inward-facing conformers of a transporter, wherein the substrate binding pocket is accessible to the extracellular or intracellular milieu, respectively, are defined as the exofacial or endofacial inhibitors[18,25]. For medicinal chemistry, a biological target that is readily accessible on the outer cell surface can avoid issues with cell penetration and open possibilities to exploit drug conjugation as in the case of the insulin-conjugated GLUT inhibitors[15]. Therefore, we have sought to focus on exofacial inhibitors.

Toward this goal, we previously engineered a variant of the xylose transporter XylE from *E. coli*, a well-studied homolog of GLUTs (Supplementary Fig. 1a), as a surrogate to screen for exofacial ligands for GLUTs[26–29]. Two residues lining the extracellular interface of the N and C domains of XylE are replaced by bulky Trp residues, resulting in a constitutively outward-facing mutant, designated XylE-WW. A combination of microscale thermophoresis (MST)-based binding assay and proteoliposome-based transport assay was performed to compare the inhibitory effects of the ligands on wide type (WT) vs. XylE-WW variant[29].

The surrogate XylE-WW worked well for several well-established GLUT inhibitors, such as the endofacial inhibitor cytochalasin B (CCB) and exofacial phloretin[29,30]. However, our ensuing studies showed discrepancies in inhibition efficiency between XylE and GLUT1/3 when more inhibitors were tested. We thereby focused on GLUT3, whose structures were captured in the outward-facing conformations at high resolutions, for engineering and structural investigation.

Here we report a GLUT3 variant, designated GLUT3exo, as a tool for screening and validating exofacial GLUT inhibitors. We determine crystal structure of GLUT3exo at 2.1 Å resolution to confirm the outward-facing conformation. Using this tool, we succeed in validating an exofacial inhibitor SA47 and solve the crystal structure of its complex with GLUT3 at 2.3 Å resolution using lipidic cubic phase (LCP) method.

## Results

**Identification and characterization of GLUT inhibitors**. Initially, a biased medium-throughput screening (MTS) was performed for novel motifs interacting with GLUT1. To this end, ~ 7000 internal compounds were selected by considering carbohydrate headgroups in analogy to glucose as natural substrate, isolated natural products, a broader carbohydrate screening collection and virtual screening results. The latter virtual screening was carried out using 2D fingerprint similarity employing known active carbohydrates and literature actives for this target. All compounds were tested for GLUT1 inhibition at a 100 μM concentration, resulting in a hit-rate of 27.5%. Threshold for confirmed hits was set to $3 * \sigma = 15.4\%$.

Backscreening for attractive clusters and singletons was carried out, followed by $IC_{50}$ determination for 300 of the most promising analogs that were selected by initial activity and chemical attractiveness. Dose-response testing resulted in a high hit-rate of 54.6% (164 compounds) and 6.7% (20 compounds) with $IC_{50}$ values < 50 μM and < 10 μM, respectively. All screened and synthesized compounds were characterized for their inhibitory potential in a transporter assay using 2-desoxy-glucose (DOG) as a non-metabolizable substrate as described earlier[31]. The $IC_{50}$ values derived from this transport assay were treated as a surrogate for evaluating compound affinity with GLUT1. The final round for prioritization was based on profiling data in addition to quality control (i.e., LCMS purity, chemical stability), the number of other targets addressed by a compound (frequent hitters) and clustering.

This screening led to identification of the 5,7-diazaindazole series as attractive series for further optimization, plus other series described in the literature[31,32]. Further optimization of an initial screening hit with a GLUT1 $IC_{50}$ value of 46.5 μM led us to explore the ortho-position at the N1-aromatic ring and to add various side chains. This effort produced SA47 as an interesting motif for further investigation, as our intention was to connect a suitable linker moiety for potentially attaching peptide conjugates.

**Engineering GLUT3 for identifying exofacial inhibitors**. When using XylE-WW to probe exofacial GLUT inhibitors, we noticed substantial distinctions in the inhibition potency and selectivity on XylE and GLUT1/3 by inhibitors SA1-SA6 from Sanofi library (Supplementary Fig. 1b, c, d, SI text)[31,33]. SA1, 2, 4, and 6 almost completely abolished glucose uptake by GLUT1/3, and SA3 mildly inhibited the GLUT1/3 transport activities. By contrast, only SA1 and SA2 moderately inhibited xylose transport by XylE.

As the inhibition pattern and potency of these inhibitors on GLUT3 is nearly identical to that on GLUT1 (Supplementary Fig. 1c, d), we focused on GLUT3 for tool development because of its higher stability after purification. Following the same principle

for the engineering of XylE-WW, we introduced two Trp substitutions, S64W and I305W, on the extracellular segments of TM2 and TM8 of GLUT3, respectively. This variant will be referred to as GLUT3exo hereafter (Fig. 1a). The two bulky substituents were expected to trap GLUT3 in an outward-facing conformation. Consistently, the transport activity of GLUT3exo was abolished as measured in proteoliposome-based counterflow assay (Fig. 1b).

To validate the conformational state of GLUT3exo, we resolved the crystal structure of GLUT3exo in complex with glucose at 2.1 Å using LCP crystallization approach (Supplementary Table 1). As expected, GLUT3exo exhibits an outward-facing state with a glucose molecule standing in its central pocket (Fig. 1c). The two bulky residues, Trp64 and Trp305, wedge into the two lateral interfaces of the N and C domains, i.e., between TM2 and TM11 and between TM5 and TM8, respectively, to lock the outward-facing conformation (Fig. 1d). Except for these two Trp substitutions, the structure of GLUT3exo is nearly identical to that of the outward-occluded GLUT3-glucose with a root-mean-square deviation (RMSD) of 0.236 Å over 402 Cα atoms (Supplementary Fig. 2a). Notably, the structures of GLUT3exo and XylE-WW structures are aligned with a larger RMSD of 1.241 Å over 310 Cα atoms (Supplementary Fig. 2b). Although the substrate coordinating residues are highly conserved between GLUT3 and XylE, a number of mismatched residues are observed in the central pocket, explaining the different sensitivities of GLUT/3 and XylE to the inhibitors (Supplementary Fig. 2c).

The feasibility of using GLUT3exo for discriminating exo- and endo-facial inhibitors was validated by microscale thermophoresis (MST) measurement with CCB and phloretin (Fig. 1e). WT GLUT3, which alternates between different conformational states, bind to CCB and phloretin with $K_D$ values of 848.5 ± 181.9 nM and 26.3 ± 3.2 μM, respectively. Consistent with its trapped outward-facing conformation, GLUT3exo can only recognize the exofacial inhibitor phloretin, with a $K_D$ of 15.5 ± 1.5 μM. Its affinity with the endofacial inhibitor CCB was not detectable (Fig. 1e).

Taken together, the introduced double Trp residues fix GLUT3exo in an outward-facing state, leaving the rest of the structure nearly identical to WT protein. Therefore, GLUT3exo can be used as a tool for examining exofacial inhibitors of GLUTs.

**Validation of SA47 as an exofacial GLUT1/3 inhibitor.** As a proof of principle, we used GLUT3exo to characterize SA47, which, modified from the GLUT1 inhibitor class 1*H*-pyrazolo[3,4-*d*]pyrimidines[33], comprises four moieties, an aryl head [1], a piperazine moiety [2], a pyrazolopyrimidine core [3], and a pyrazoloaryl tail [4] (Fig. 2a).

Transport of ³H-glucose by GLUT1 and GLUT3 was inhibited by SA47 with IC₅₀ of 331.4 ± 38.3 nM and 822.6 ± 138.3 nM, respectively (Fig. 2b). Consistently, the affinity of SA47 with GLUT1 and GLUT3 was measured with $K_D$ values of 309.5 ± 58.2 nM and 316.5 ± 47.2 nM, respectively, using MST (Fig. 2c). Importantly, SA47 binds to GLUT3exo with a $K_D$ of 313.6 ± 80.7 nM, suggesting it to be an exofacial GLUT inhibitor (Fig. 2c).

To elucidate the mode of action (MOA) of SA47, we determined the crystal structure of SA47-bound WT GLUT3 (GLUT3-SA47) complex at 2.3 Å resolution (Fig. 2d and Supplementary Table 1). SA47 is unambiguously resolved in the central cavity, which opens to the extracellular side, consistent with the predicted outward-open conformation (Fig. 2d, e). Scrutiny of the electron density of SA47 reveals two conformers with regard to its pyrazoloaryl tail [4]. The tail stretches towards the extracellular side in Conformer 1 and bends to be parallel with the membrane plane in Conformer 2 (Fig. 2e).

**SA47 blocks the extracellular access to the substrate binding pocket.** Calculation of the substrate transport path using HOLE[34] shows that SA47-bound GLUT3 exhibits a similar conformation with the maltose-bound GLUT3 structure (PDB code: 4ZWC). Structural comparison of outward-open GLUT3-SA47 to outward-occluded GLUT3-glucose (PDB code: 4ZWB) reveals partial overlap between the bound SA47 and glucose molecules (Fig. 3a, b). The piperazine moiety [2] and the pyrazolopyrimidine core [3] of SA47 partially overlay with the extracellular half of the glucose molecule. The aryl head [1] stands aside the 3-hydroxy group of glucose, and the pyrazoloaryl tail [4] projects to the extracellular side along the open tunnel. The comparison unambiguously shows the inhibitory mechanism of SA47 to be blocking substrate access to the central binding site.

To widen the extracellular tunnel, TM2/7b/9/12 and the extracellular helix TM1e move away from the central path (Fig. 3c). Among these, TM7b undergoes the most pronounced shift with a combination of overall motion, helix bending, and axial rotation (Fig. 3c). The structural shifts result in the displacement of the extracellular gating residue Tyr290, completing the transition from outward-occluded to outward-open (Fig. 3d).

**Interaction between SA47 and GLUT3.** SA47 is coordinated by GLUT3 through extensive interactions, including hydrogen bonds (H-bonds), π stacking, and van der Waals contacts (Fig. 4a). We will depict the detailed coordination of each of the four moieties in SA47 (Fig. 4b).

The aryl head [1] of SA47 resides in the central cavity of GLUT3 encompassed by TM1, TM2, TM7, and TM11, coordinated via both polar and hydrophobic interactions (Fig. 4a). Thr28 on TM1 is H-bonded to the fluorine of SA47. Another H bond is formed between Gln281 on TM7 and the methoxy group of SA47, which is stabilized by the H-bond network among Ser71-Asn413-Gln281-Asn409-Gln280. Three nearby bulk residues, Phe24 on TM1, Phe70 on TM2, and Trp410 on TM11, as well as Trp386 on TM10 form hydrophobic contacts with the aryl head [1] (Fig. 4b, top).

The piperazine moiety [2] has contact through hydrophobic interaction with Trp386 on TM10, and the nitrogen atom adjacent to ring [1] is H-bonded to Gln281 on TM7. The pyrazolopyrimidine moiety [3] is stably bound via polar interactions with Gln159 on TM5, Asn315 on TM8, and Glu378 on TM8 and π-π stacking with Phe377 (Fig. 4b, bottom).

While the benzyl group of the pyrazoloaryl tail [4] forms a π-π stacking with Phe289 on TM7b, the two conformers of the aryl tail are coordinated differently by GLUT3 residues (Fig. 4a, c). The stretching conformer 1 forms three hydrogen bonds with Asn32 on TM1 and Gln170 on TM5 (Fig. 4c, top). The transverse conformer 2 still interacts with Asn32, but is out of the reach of Gln170 (Fig. 4c, bottom).

To validate the structure-revealed coordination of SA47, we performed systematic Ala substitution for ligand binding residues on GLUT3. Among the tested mutants, expression of mutants containing the single point mutation Q170A, Q280A, Q281A, F377A, and N409A could not be detected, and that of F24A or W410A was substantially reduced, consistent with the structural role of these residues. We focused on the single point mutants whose protein yield and behavior is similar to WT for MST measurement (Fig. 4d and Supplementary Table 2).

Supporting the structural observation, Ala replacement of most of the SA47-coordinating residues led to reduced affinities to different degrees (Fig. 4d and Supplementary Table 2). The only exception was F289A, which exhibited an enhanced affinity to 163.8 ± 67.0 nM. As Phe289 is positioned to the breaking point of

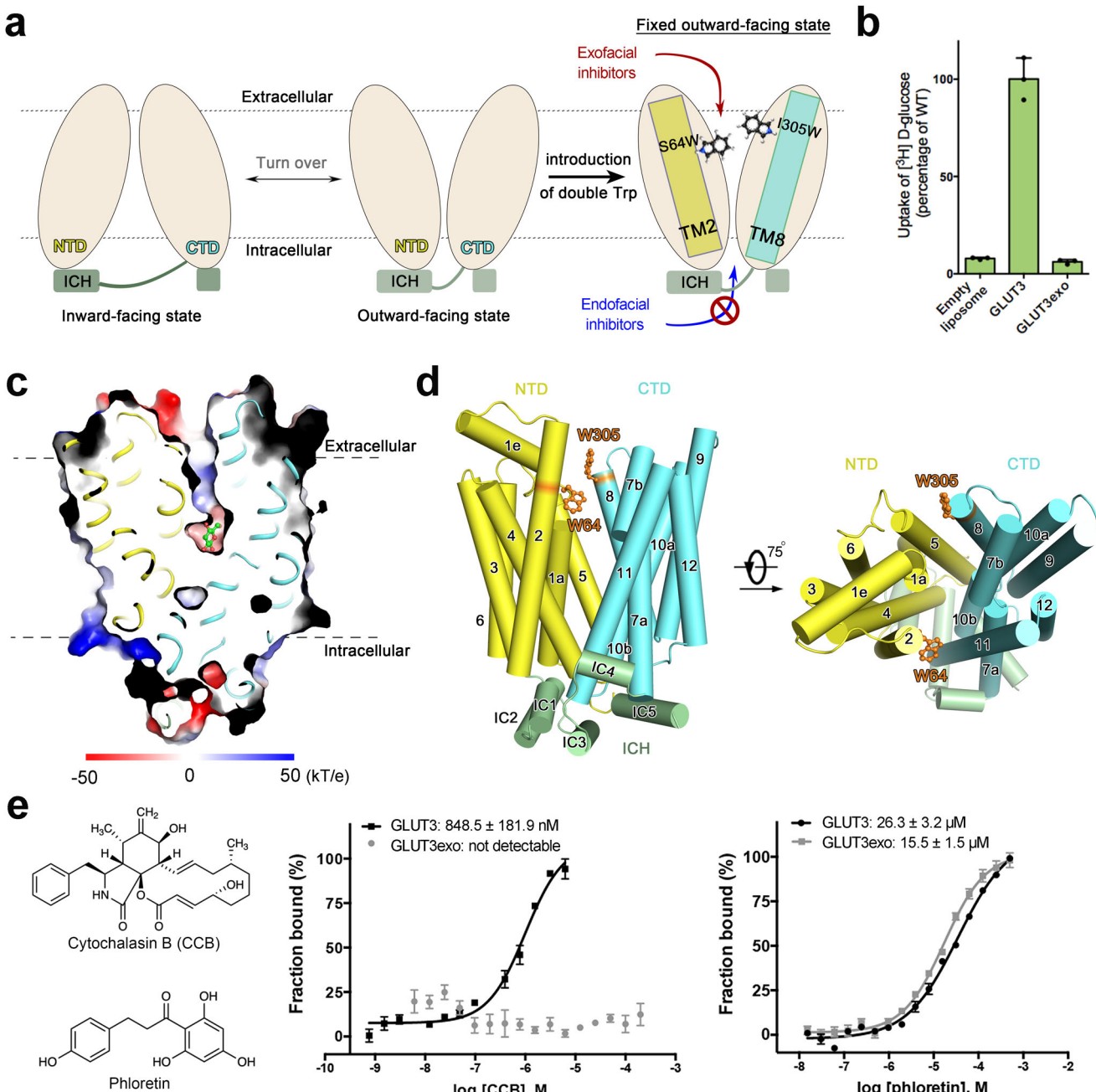

**Fig. 1 GLUT3 containing a double point mutation is trapped in an outward-facing state. a** Structure-guided design of a constitutively outward-facing GLUT3 variant. Wild-type (WT) GLUT3 alternates between the inward-facing (left) and the outward-facing (middle) states during a transport cycle. Introduction of a double Trp mutation, S64W and I305W, designated GLUT3exo, may lock the transporter in a constitutively outward-facing state that can only bind to exofacial inhibitors. NTD amino-terminal domain, CTD carboxy-terminal domain, ICH intracellular helical domain, TM transmembrane segment. **b** GLUT3exo loses transport activity. The activities of GLUT3 variants were examined in the proteoliposome-based counterflow assay. Empty liposome refers to protein-free control. The transport activity of GLUT3exo was normalized against that of WT GLUT3. Data represent mean ± SD of three independent experiments. **c** Crystal structure of GLUT3exo displays an outward-facing state. Shown here is a cut-open side view of the electrostatic surface potential of GLUT3exo. Glucose is shown as green ball and sticks. **d** Overall structure of the outward-facing GLUT3exo. The introduced Trp residues are shown as orange ball and sticks. GLUT3exo is domain colored with yellow, cyan, and pale green for the N-terminal, C-terminal, and ICH domains, respectively. **e** GLUT3exo selectively binds to the exofacial inhibitor phloretin, but not the endofacial cytochalasin B (CCB). *Left*: Chemical structures of CCB and phloretin. *Middle* and *Right*: Microscale thermophoresis (MST) measurement of the affinities of WT GLUT3 and GLUT3exo with the endofacial inhibitor CCB (middle) and the exofacial inhibitor phloretin (right). Data represent mean ± SEM of three independent measurements. Source data are provided as a Source Data file.

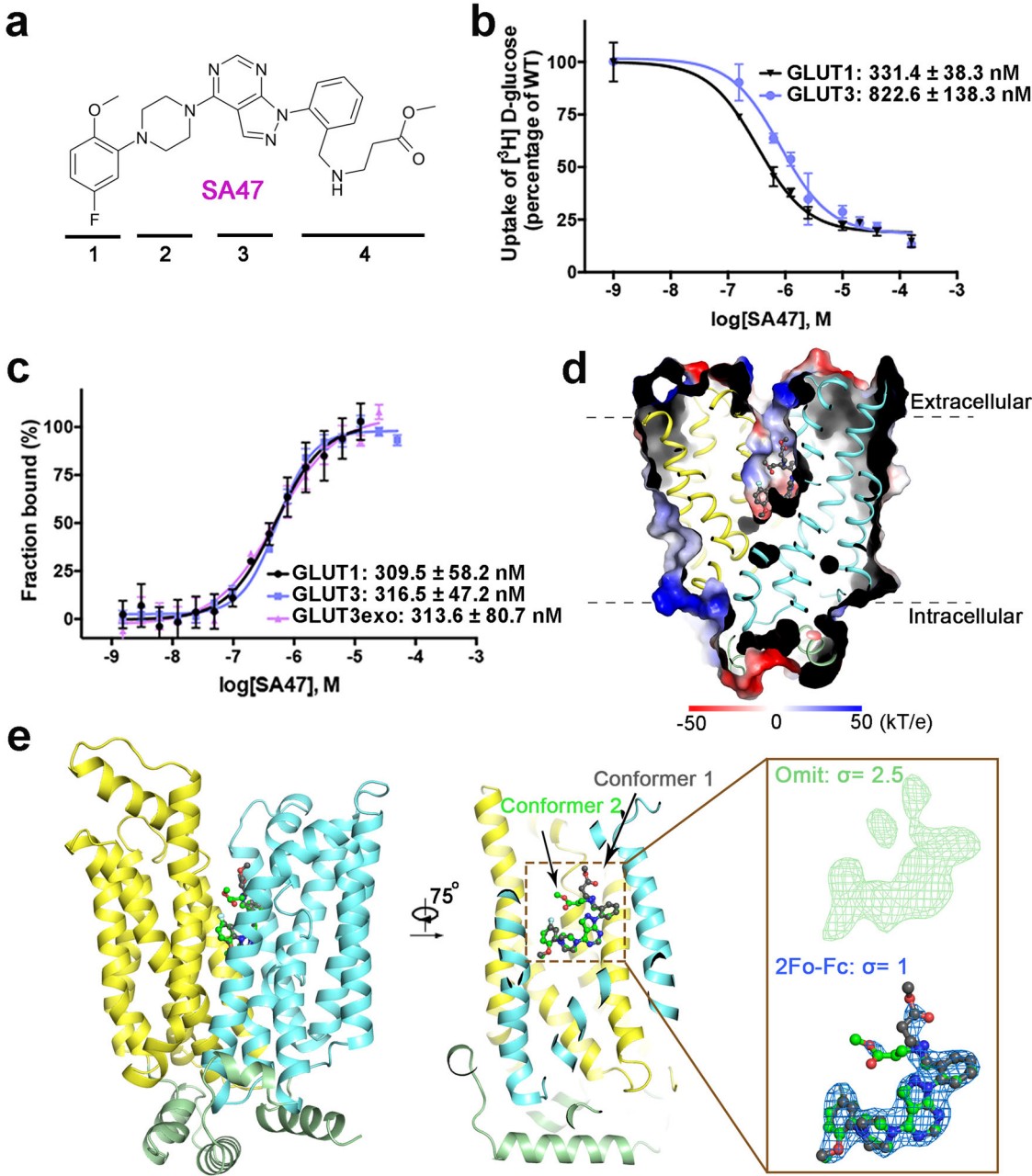

**Fig. 2 SA47 is an exofacial GLUT inhibitor. a** Chemical structure of SA47. The four moieties of SA47 are indicated by the numbers below the structure. **b** Inhibition of GLUT1 and GLUT3 by SA47. Data represents mean ± SD of three independent experiments. **c** SA47 has similar affinity with GLUT1, GLUT3, and GLUT3exo. Data represent mean ± SEM of three independent MST measurements. **d** Crystal structure of SA47-bound GLUT3 (GLUT3-SA47) in an outward-facing state. Shown here is the cut-open side view of the electrostatic surface potential of GLUT3-SA47. **e** Binding pose of SA47. *Inset*: Electron density of SA47. The omit map and 2Fo-Fc maps are shown as green and blue meshes, contoured at 2.5 σ and 1 σ, respectively. Two conformers of SA47, shown as ball and sticks, are colored dark gray and green. Source data are provided as a Source Data file.

TM7b in the outward-open conformation (Fig. 3d), substitution of the bulky aromatic ring may lower the energy penalty for the conformational change that is required for accommodating SA47, hence resulting in an increase in the nominal affinity.

## Discussion

GLUTs have been rigorously investigated for decades owing to their fundamental role to maintain the cellular glucose homeostasis. Elevated expression of GLUT isoforms in advanced cancer cells has opened the possibility of exploiting GLUT inhibitors for cancer treatment. To date, more than ten classes of potent GLUT inhibitors have been developed through high-throughput screening (HTS) and the structure–activity-relationship (SAR)-based optimization of small molecules[35–37]. Prior to this study, the only structures of GLUTs with exofacial ligands are GLUT3 bound to maltose in the outward-open and occluded states. Maltose is a physiologically abundant disaccharide that can weakly inhibit glucose transport by GLUT1 and GLUT3. In contrast to the sub-micromolar range IC$_{50}$ of SA47, maltose cannot completely inhibit GLUT1 and GLUT3 even when applied at 50 mM[18].

Structures of GLUT3-SA47 and GLUT3-maltose can be superimposed with the RMSD of 0.452 Å over 418 Cα atoms (Fig. 5a). Most of the substrate binding residues share similar conformations

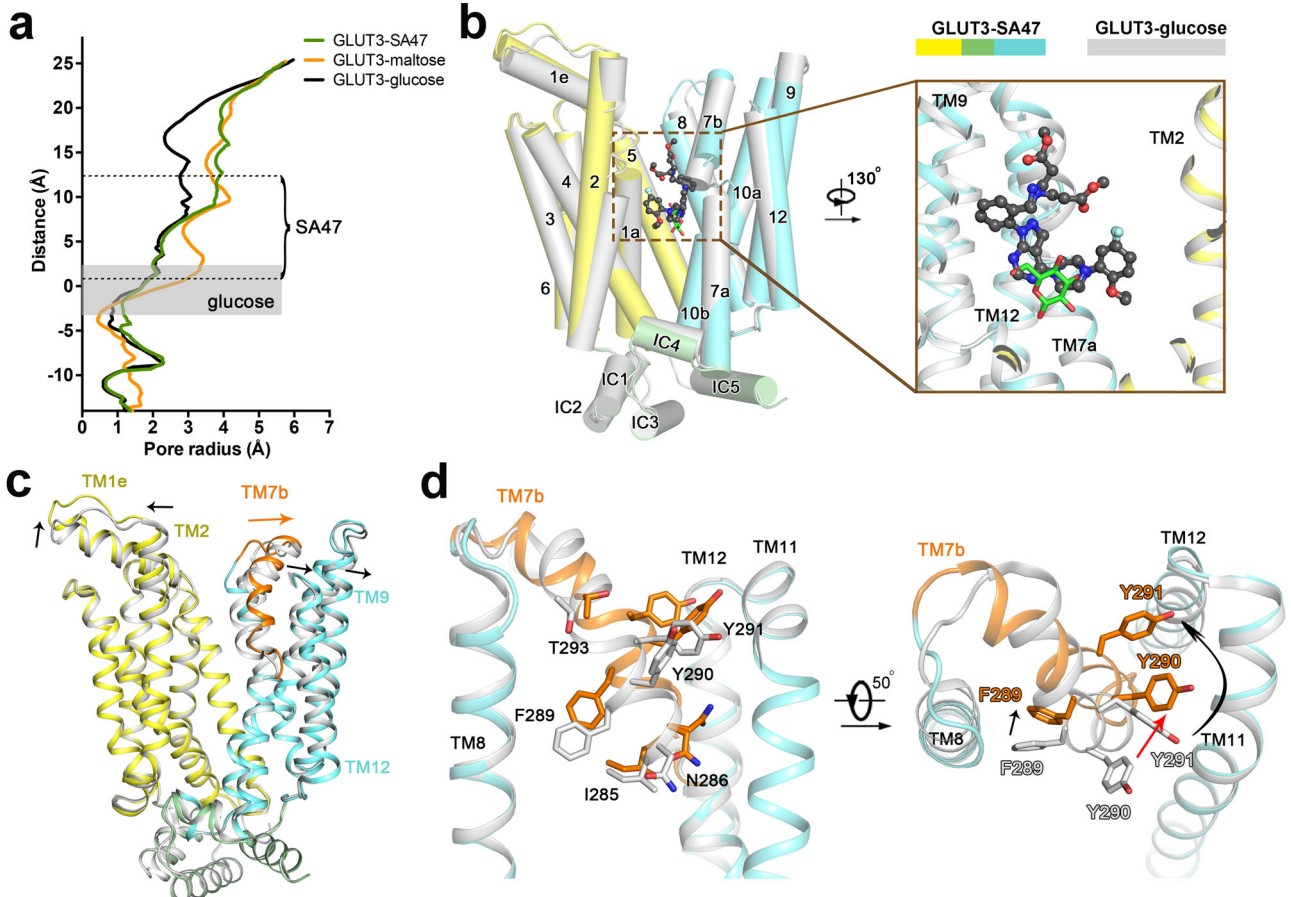

**Fig. 3 Structural comparison of GLUT3-SA47 and GLUT3-glucose complexes. a** The structure of GLUT3-SA47 exhibits a similar outward-open state as that of maltose-bound GLUT3. The radii of extracellular substrate access tunnel in the indicated GLUT3 structures were calculated in HOLE[34]. The glucose-binding site in GLUT3-glucose complex is indicated by the gray shade. **b** Partial overlap between SA47 and glucose in GLUT3. TM11 is omitted for a clear view of ligands. SA47 and glucose are shown as black ball-and-sticks and green sticks, respectively. **c** Local conformational shifts between GLUT3-SA47 and GLUT3-glucose. The arrows indicate the TM movements that widen the path to the central pocket. TM11 is omitted for a clear view of the movement of TM7b, which is colored in GLUT3-SA47. **d** Structural shift of TM7b residues upon SA47 binding. The corresponding TM7b residues in GLUT3-SA47 and GLUT3-glucose are colored orange and gray, respectively.

between GLUT3-SA47 and GLUT3-maltose (Fig. 5b). The major deviation occurs to the local shift of Phe289 (Fig. 5b). Compared to GLUT3-maltose, the aromatic ring of Phe289 on TM7b rotates away from the central transport path because of its potential steric clash with the benzyl group of pyrazoloaryl tail [4]. This may in part explain the increased affinity with SA47 when Phe289 was substituted with Ala (Fig. 4c). A minor difference is found in the swing of Asn286. To avoid steric hindrance with the the second glucose unit (Glc2) moiety of maltose, the side chain of Asn286 points away from the central cavity. In the SA47-bound structure, Asn286 exists in two conformers because of the lack of hindrance with the bound compound (Fig. 5b).

The drastic difference of the chemical structures of maltose and SA47 versus the similar structures of GLUT3 in the outward-open state demonstrates the tolerance of the extracellular transport path for different chemicals. This half tunnel thus represents a potential site for drug discovery. Supporting this notion, GLUT3 can also bind to a weak inhibitor C3361, which is selective for PfHT1 (Supplementary Fig. 3)[38]. The linear tail of C3361, which points to different directions in GLUT3 and PfHT1[16,39], was derived to fill in the exofacial pocket of PfHT1, resulting in more potent inhibitors of PfHT1 with increased selectivity and reduced toxicity[16].

The N and C domains of GLUTs undergo rotation to achieve alternating access. The intracellular cavity in the inward-open structures is completely different from that in the outward-facing ones. Three GLUT1 structures in complex with endofacial inhibitors, including CCB, GLUTi1, and GLUTi2, have been elucidated[24]. Structural comparison shows that the binding pockets for exofacial and endofacial inhibitors are independently located on the opposite sides of the glucose-binding pocket (Fig. 5c). This discovery highlights the importance of obtaining structures of GLUTs in complex with different types of inhibitors to facilitate drug discovery.

Taken together, we established and optimized a reliable tool to select exofacial GLUT inhibitors and validated SA47 as a potent exofacial GLUT inhibitor. The aryl tail [4] of SA47, which points to the extracellular transport path in GLUT3, can be conveniently linked with insulin to generate a local depot as a glucose-controlled insulin-delivery system[15]. By attaining the high-resolution structure of GLUT3-SA47 complex, we elucidate the MOA of the pyrazolopyrimidine inhibitor, which lays the foundation for further optimization of exofacial GLUT inhibitors.

## Methods

**Protein expression and purification.** The recombinant human glucose transporter GLUT1(N45T) and *E. coli* (strain O157:H7) D-xylose:proton symporter XylE were expressed and purified as described previously[17,26]. The expression and purification of human GLUT3(N43T), wild type and mutants, was performed using

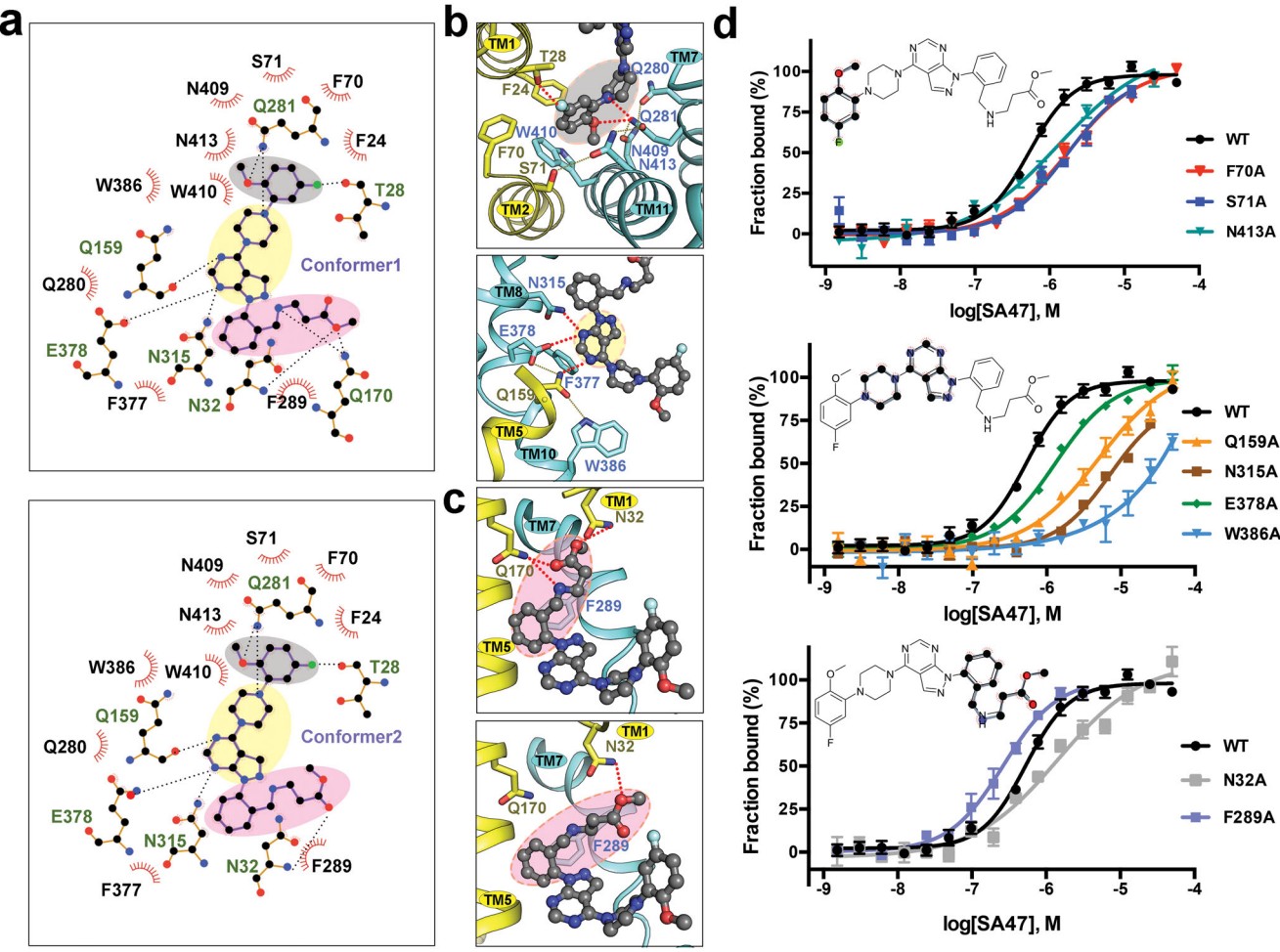

**Fig. 4 Coordination of SA47 by GLUT3. a** Schematic illustration of the interactions between GLUT3 and the two conformers of SA47. Schematics were generated in LIGPLOT + v.1.4.5. H-bonds are indicated by black, dashed lines. Hydrophobic contacts and π stacking interactions are represented by eyelash motifs. **b** Coordination of the first three moieties of SA47. *Top*: coordination of the aryl head [1] (gray shade). *Bottom*: Coordination of the piperazine moiety [2] and the pyrazolopyrimidine core [3] (wheat shade). **c** Coordination of the two conformers of the pyrazoloaryl tail [4] (pink shade). Hydrogen bonds are indicated by red dashed lines. **d** Biochemical validation of the SA47 binding site. MST were performed to measure the interactions between SA47 and indicated GLUT3 variants. From top to bottom, GLUT3 mutants that were designed to impair the coordination of the aryl head [1] (top), the piperazine moiety [2] and the pyrazolopyrimidine core [3] (middle), and the pyrazoloaryl tail [4] (bottom) of SA47 were examined. Fractions bound of GLUT3 variants were normalized against that of wild-type GLUT3. Data represent mean ± SEM of three independent measurements. Please refer to Supplementary Table 2 for the summary of $K_D$ of the GLUT3 mutants.

a similar protocol with minor modifications[18]. In brief, the cDNA of human GLUT3 was cloned into a modified pFastBac vector (Invitrogen) fused with an N-terminal His$_{10}$ tag. All mutants were generated by standard PCR-based strategy.

Following the instructions for the Bac-to-Bac baculovirus expression system (Invitrogen), proteins are overexpressed in the Sf9 insect cells with infected baculovirus after viral infection of 72 h. The protein was solubilized from Sf9 insect cell membrane with 2% (w/v) n-dodecyl-β-D-maltoside (DDM, Anatrace) at 4 °C for 1.5 h and purified using nickel affinity resin (Ni-NTA, Qiagen). The protein was washed with the buffer containing 25 mM 2-(N-morpholino)ethanesulfonic acid (MES) pH 6.0, 150 mM NaCl, 30 mM imidazole, and 0.06% (w/v) 6-cyclohexyl-1-hexyl-b-D-maltoside (CYMAL-6, Anatrace) and eluted with the wash buffer plus 270 mM imidazole. Concentrated protein was applied to the size-exclusion chromatography (SEC) (Superdex-200, GE Healthcare) pre-equilibrated with the buffer containing 25 mM MES pH 6.0, 150 mM NaCl, and 0.06% (w/v) CYMAL-6. Peak fractions were collected and concentrated to approximately 40 mg/ml using an Amicon Ultra 50 K filter (Millipore) for crystallization trials.

**Chemical synthesis.** All screened and synthesized compounds were characterized for their inhibitory potential in a transporter assay using 2-desoxy-glucose (DOG) as a non-metabolizable substrate as described in the literature[31]. The origin and synthesis routes of the compounds are detailed in Supplementary Note 1.

**Crystallization.** Both glucose-bound GLUT3exo and GLUT3-SA47 were crystallized using LCP. For the GLUT3exo-glucose complex, concentrated GLUT3exo was mixed

with D-glucose at a final concentration of 50 mM at 4 °C for 30 min. Then the protein was mixed with monoolein (Sigma) with a protein to lipid ration (w/w) of approximately 2:3 using a syringe lipid mixer. For crystallization, 45 nl meso phase solution was mixed with 1000 nl crystallization buffer containing 40% (v/v) PEG400, 100 mM Tris pH 7.6, and 100 mM MgCl$_2$ on the glass sandwich plates.

For the GLUT3-SA47 complex, concentrated GLUT3 protein was incubated with 5 mM SA47 on ice for 30 min. Following the aforementioned protocol for mesophase formation, 45 nl mesophase was dispensed on a glass plate and then overlaid by 1000 nl crystallization buffer consisting of 40% (v/v) PEG400, 100 mM sodium citrate pH 5.7, and 400 mM potassium formate on glass sandwich plates.

All LCP crystals grew to full size at 20 °C within one week. The crystals were collected using MicroMesh (M3-L18SP-50; MiTeGen) and flash frozen in liquid nitrogen.

**Data collection and structure determination.** The data sets of GLUT3exo-glucose and GLUT3-SA47 complexes were collected at SPring-8 microfocus beamline BL32XU, using a 10 μm × 15 μm microfocus beam with 1.0 Å wavelength. Every single crystal was diffracted for 10° with a 0.1° oscillation angle. The LCP crystals of GLUT3 were screened and automatically collected through ZOO[40] and the diffraction data sets were automatically processed by KAMO[41]. Manual data processing was applied to identify and merge the data sets with good diffraction and low R-merge factor using XDS[42]. Further processing was carried out with programs from the CCP4 suite[43]. Phase was solved by molecular replacement using PHASER[44] with the structure of GLUT3 (PDB code: 4ZW9) as a searching model.

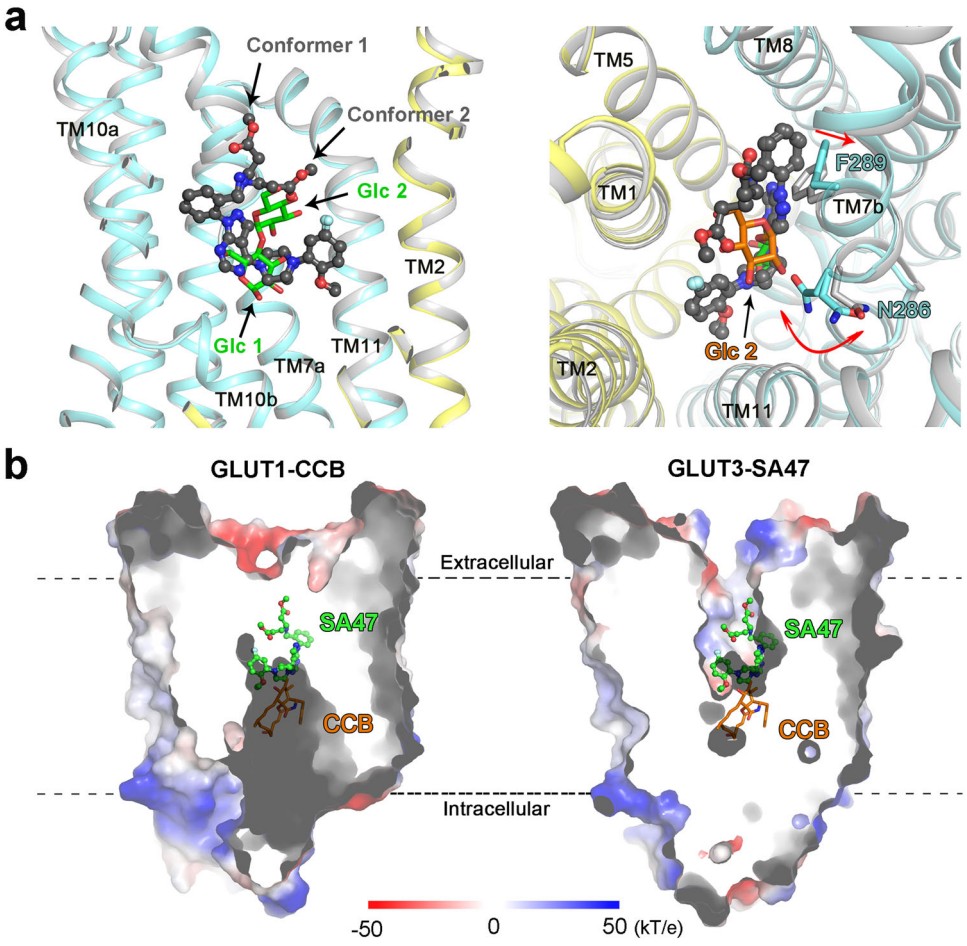

**Fig. 5 Structural comparison of different GLUT-inhibitor complexes. a** Structural comparison of GLUT3 bound to SA47 and maltose. Left: The NTD and CTD of GLUT3-SA47 were colored yellow and cyan, respectively. GLUT3-maltose structure (PDB code: 4ZWC) was colored gray. Despite the much larger size of SA47 than maltose, the overall structure of GLUT3 remains nearly unchanged in the presence of these two inhibitors. The SA47 was shown as ball-and-sticks and colored gray. The maltose was shown as stick and colored green. Glc1: the first glucose unit; Glc2: the second glucose unit. Right: Residue shifts of TM7b upon SA47 binding. Phe289 and Asn286 on TM7b undergo minor shifts between the two structures. Shown here is an extracellular view of the superimposed structures of GLUT3 bound to SA47 (domain colored) and maltose (gray, PDB code: 4ZWC). The glc2 moiety of maltose is highlighted by colored orange. **b** Structural comparison of the outward-facing GLUT3-SA47 and inward-facing GLUT1-CCB. SA47 and CCB were positioned based on overall structural superposition of GLUT1-CCB (PDB code: 5EQI) and GLUT3-SA47.

The model was rebuilt in COOT[45] and refined with PHENIX[46]. Data collection and structure refinement statistics are summarized in Supplementary Table S1.

**Preparation of liposomes and proteoliposomes.** *E. coli* polar lipids (Avanti) were dissolved, dried, and resuspended to a final concentration of 20 mg/ml with the KPM 6.5 buffer (50 mM potassium phosphate, 2 mM $MgSO_4$; pH 6.5) supplemented with 50 mM D-glucose or xylose (Sigma). After freeze-and-thaw with liquid nitrogen, the liposomes were extruded through 0.4 μm polycarbonate membranes (Millipore) and incubated with 1% n-octyl-β-d-glucoside (β-OG; Anatrace) at 4 °C for 30 min. Purified protein was incubated with liposomes at a ratio of ~1:100 (w/w) for 60 min. Then β-OG was removed by incubating with Bio-Beads SM2 (Bio-Rad) overnight. The proteoliposomes were subject to freeze-and-thaw again and extruded through 0.4 μm polycarbonate membranes. The homogenized proteoliposomes were ultracentrifuged at 100,000 g at 4 °C for 1 h and rinsed twice with the ice cold KPM 6.5 buffer. Finally, the proteoliposomes were resuspended in the KPM 6.5 buffer to a final concentration of 100 mg/ml (phospholipids).

**In vitro counterflow assay.** All counterflow assays were performed at 25 °C. For each assay, an aliquot of 2 μl proteoliposomes was added into 98 μl KPM 6.5 buffer plus 1 μCi D-[2-³H] glucose (specific radioactivity 23.4 Ci/mmol, PerkinElmer) or D-[³H] xylose (20 Ci/mmol, American Radiolabeled Chemicals, Inc.). The final concentrations of the external D-[2-³H] glucose and D-[³H] xylose were 0.42 μM and 0.5 μM, respectively. Uptake of radio-labeled substrates was terminated at 30 s by rapidly filtering the solution through 0.22 μm GSTF filters (Millipore) and washed with 2.5 ml KPM 6.5 buffer. The filter was solubilized with 0.5 ml Optiphase HISAFE

3 (PerkinElmer) for 30 min for liquid scintillation counting with MicroBeta JET (PerkinElmer). Liposome without protein was used as negative control.

For inhibition assays, an aliquot of 2 μL proteoliposome pre-incubated with 100 μM indicated inhibitors for 0.5 h on ice was added to 98 μL KPM buffer containing 1 μCi D-[2-³H]-glucose or D-[³H] xylose with 100 μM inhibitors. To determine the IC50 of the inhibitor, a series of concentrations of the inhibitor was applied as indicated.

All experiments were repeated at least three times. Data for liposome-based counterflow assay were analyzed using GraphPad Prism 6.

**Microscale thermophoresis binding assay.** Purified GLUT3 proteins, WT or mutants, were diluted in the buffer containing 25 mM MES 6.0, 150 mM NaCl, and 0.06% (w/v) CYMAL-6. Purified GLUT1 was diluted with the buffer containing 25 mM MES 6.0, 150 mM NaCl, and 0.02% (w/v) DDM. Inhibitors with serial dilutions were mixed with indicated protein. The mixture was loaded to MO-Z002 capillaries at room temperature. MST analyses were conducted on a Monolith NT. Label-free instrument (NanoTemper Technologies GmbH) with 20% LED power and 60% MST power. The MST data was analyzed by Monolith NT.115.

**Reporting summary.** Further information on research design is available in the Nature Research Reporting Summary linked to this article.

## Data availability
The coordinates and structure factors for GLUT3exo bound to glucose and GLUT3 bound to SA47 have been deposited in the Protein Data Bank (PDB, http://www.wwpdb.org)

with the accession codes 7SPT (GLUT3exo bound to glucose) and 7SPS (GLUT3 bound to SA47). The source data underlying Figs. 1b, 1e, 2b, 2c and Fig. S1b–d are provided as a Source Data file. All other data are available from the authors on request.

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

## Acknowledgements

We thank Kunio Hirata at Super Photon ring-8 (SPring-8, Japan) BL32XU beamline for the help of crystal screening and data collection. We thank the X-ray Crystallography Platform of the Tsinghua University Technology Center for Protein Research for the crystallization facility. This work was funded by the National Natural Science Foundation of China (SQ2020YFA050052, 31630017, 81861138009, and 31611130036) and Beijing Nova Program (Z201100006820039).

## Author contributions

N.Y. and X.J. conceived the project. X.J., N.W., Y.Y., and S.Z. designed the experiment. N.W., X.J., Y.Y., S.Z., and H.X. performed the experiments of purification, counterflow, and MST assays. X.J., N.W., and Y.Y., performed experiments for structural determination. E.D., H.M., M.B., V.D., M.D., N.H., K.H.H., S.P., M.P., and N.T. synthesized the GLUT inhibitors. All authors contributed to data analysis. X.J., N.W., and N.Y. wrote the manuscript.

## Competing interests

E.D., H.M., M.B., V.D., M.D., N.H., K.H., S.P., and M.P. are Sanofi employees and may hold shares and/or stock options in the company. The remaining authors declare no competing interests.
