## [Peer Review File · Nature Communications]

Molecular basis for inhibiting human glucose transporters by exofacial inhibitorsEditorial Note: Parts of this Peer Review File have been redacted as indicated to remove third party material where no permission to publish were obtained

Reviewers' comments:

Reviewer #1 (Remarks to the Author):

In this manuscript, Wang et al rationally engineered the human glucose transporter GLUT3 to lock its conformation as outward-facing (GLUT3_{exo}) for inhibitor screening, biochemically and structurally validated the design, obtained the inhibitor SA47 that is chemically distinct from the substrates, and illustrated the inhibition mechanism by structural characterization and mutagenesis. Although the anticancer potential of the chemicals is unclear from the current study, and the validity of GLUT1/3 as anticancer drug targets remains to be tested, the work presents a strategy for the rational design of inhibitors of transporters that may be generally useful. The manuscript is easy to understand and the presentation is clear in general.

The major aim of the study is to develop GLUT1/3-specific inhibitors but the specificity of SA47 for GLUTs was not reported. Given the high similarity between GLUT1/3 to GLUT2/4, the inhibition of SA47 on GLUT2/4 should be tested or at least discussed.

Some minor comments for the authors to consider

Line 95/96, no data are shown for the comparison between XylE and the WW variant.

Line 98/99, the authors should elaborate on what the discrepancies are in the main text.

Line 130, what was the reason for the weakened affinity of GLUT3_{exo} compared with WT for phloretin?

Line 227, could the authors elaborate on what “chemical property” of maltose is not suitable for optimization?

Line 285, what was the membrane cut-off for the concentrator?

Line 296/301, mesophase is not a solution; and the mesophase is not “mixed” with the precipitant solution.

Line 324, please cite the membrane size for extrusion.

Line 325, please specify the ratio of protein:lipids (molarity or mass, for example).

Reviewer #2 (Remarks to the Author):

The paper under review essentially presents a structural biology investigation of the GLUT3 transporter clad in the disguise of drug discovery.

The authors argue that it will be advantageous to develop exofacial GLUT inhibitors, but, in fact, the developments of the last decade have yielded diverse compound classes with cellular and in vivo activity demonstrating that GLUT inhibitors can be found readily, that appear to penetrate cells and tissues. This may not have been proven explicitly, but then the authors also did not prove that the compounds they identified hit the protein from the outside. For instance, in the GPCR field ant/agonists have been developed that penetrate membranes first and then engage their targets by lateral diffusion.

This argument set aside there are several major weaknesses in the paper:

1. The authors claim that their mutant can be used to identify novel inhibitors and refer to a “biased screening of the Sanofi library”.

What was the screening set-up, and which readout has been used? Dynamic range, Z'-factor, hit rate and further data are not given. It is unclear whether the mutant can indeed be used for screening, the claim is not substantiated.

2. The authors write that their structure can be used for drug design. This may, in principle, be so, but for GLUT-inhibition, dual GLUT1/3-inhibitors are required (also stressed by the authors). So, how can the current data set be used to devise dual inhibitors? Again, there is a far-reaching claim, but no proof – notably not even for improvement of the identified inhibitors. It would actually be doable to validate for the Bayer compound class which the authors rediscovered because Bayer published a large data set, and the paper contains data for GLUT3. Can structure-design based on the current structure explain these data?

3. Despite the fact that the authors claim use of their mutant for identification of new inhibitors, the paper does not contain a new class that could be pursued with faith (compounds SA1-5 do not really qualify).

4. There are several GLUT structures published by now and both GLUT1 and -3 have to be considered here (see above; GLUT1-structures with ligands in, are also known, a Bayer paper). This novelty of the structure shown here is limited, and since drug design will need the compound bound to both GLUT1 and -3 (see above), applicability is limited as well. In addition, the strategy to lock transporters into the outward-open conformation by appropriate mutations has been published before by the authors (as they also write), such that this is not novel methodology.

Taken together, the paper under review reports an exo-open crystal structure of GLUT3 with a previously reported inhibitor chemotype bond. This finding as such should indeed be reported to the community but not in a Nature Communications paper.

I recommend rejection.

Reviewer #3 (Remarks to the Author):

In this manuscript by Wang et al., the authors have investigated the molecular basis for inhibiting human GLUTs, in particular they have focused on GLUT3. This is highly relevant study and it should be published. The three-dimensional structure of GLUT3 in complex with a novel inhibitor was determined and the activity was confirmed by uptake measurement and the binding affinity was also determined. The manuscript is well-written and easy to follow.

1. In general, I believe this a solid manuscript, what is lacking is, however, the uptake measurement in wtGLUT3 with the SA47 inhibitor in a comparison with maltose. The authors state that the inhibitor (SA47) binds to GLUT3 in a similar manner as maltose, but since maltose is not suitable for lead optimization, this manuscript is still of clear importance. However, the authors do not show how well the glucose uptake is blocked by SA47 compared to maltose. Thus, the author needs to include maltose as control in Fig 2b.

2. Figure 5 is labelled wrongly, the figure legend does not fit the figure, for instance there is "c" in the legend that is not in the figure.

Minors:

1. The authors are using mixed code for residues, use one or three letter code consistently in the text and figures.

Response to Reviewers:

Reviewer #1:

This reviewer commented that “Wang et al rationally engineered the human glucose transporter GLUT3 to lock its conformation as outward-facing (GLUT3_{exo}) for inhibitor screening, biochemically and structurally validated the design, obtained the inhibitor SA47 that is chemically distinct from the substrates, and illustrated the inhibition mechanism by structural characterization and mutagenesis. Although the anticancer potential of the chemicals is unclear from the current study, and the validity of GLUT1/3 as anticancer drug targets remains to be tested, the work presents a strategy for the rational design of inhibitors of transporters that may be generally useful.”

This reviewer raised several specific points that are addressed below:

Major points:

1. The major aim of the study is to develop GLUT1/3-specific inhibitors but the specificity of SA47 for GLUTs was not reported. Given the high similarity between GLUT1/3 to GLUT2/4, the inhibition of SA47 on GLUT2/4 should be tested or at least discussed.

We appreciate this insightful question. Given that recombinant expression and purification of GLUT2/4 was technically difficult, we did not measure the potency of SA47 against GLUT2/4. Nonetheless, our results afford clue to the subtype-specific sensitivity of SA47. As shown below, all residues involved in SA47 binding are identical in GLUT1 and GLUT3, but not GLUT2/4, wherein the locus corresponding to Thr28 is replaced by Ile. In addition, the corresponding residue of Ser71 in GLUT2 is Ala. T28A abolished SA47 binding to GLUT3, and S71A decreased the affinity with SA47. These data suggest that SA47 should have a higher potency for GLUT1/3 than GLUT2/4. Supporting this analysis, a close derivative of SA47, **compd 15**, exhibited selective inhibition to GLUT1/3 over GLUT2 (Siebeneicher et al., 2016).

	24	28	32	70	71	159	170	280	281	286	289	315	377	378	386	409	410	413
GLUT1	F	T	N	F	S	Q	Q	Q	Q	N	F	N	F	E	W	N	W	N
GLUT2	F	I	N	F	A	Q	Q	Q	Q	N	F	N	F	E	W	N	W	N
GLUT3	F	T	N	F	S	Q	Q	Q	Q	N	F	N	F	E	W	N	W	N
GLUT4	F	I	N	F	S	Q	Q	Q	Q	N	F	N	F	E	W	N	W	N

Minor points:

2. Line 95/96, no data are shown for the comparison between XylE and the WW variant.

The comparison between XylE-WT and XylE-WW, shown below, has been published in our previous paper (Jiang et al., 2019), which was cited in the present manuscript (Ref 29). [Redacted]

3. Line 98/99, the authors should elaborate on what the discrepancies are in the main text.

Point taken. Discrepancies of inhibition potency between XylE and GLUT1/3 have been elaborated in the revised manuscript “*When using XylE-WW to probe exofacial GLUT inhibitors, we noticed substantial distinctions in the inhibition potency and selectivity on XylE and GLUT1/3 by inhibitors SA1-SA6 from Sanofi library (Supplementary Fig. 1b, c, d, SI text)*”^{31,33}. SA1, 2, 4, and 6 almost completely abolished glucose uptake by GLUT1/3, and SA3 mildly inhibited the GLUT1/3 transport activities. By contrast, only SA1 and SA2 moderately inhibited xylose transport by XylE.” (Page 6 in the revised manuscript).

4. Line 130, what was the reason for the weakened affinity of GLUT3_{exo} compared with WT for phloretin?

The affinity of GLUT3_{exo} is not weakened. The dissociation constant (K_d) is inversely proportional to the affinity (or association constant, K_a). In our study, phloretin has a higher affinity with GLUT3_{exo} ($K_d = 15.5 \pm 1.5 \mu\text{M}$) than with WT GLUT3 ($K_d = 26.3 \pm 3.2 \mu\text{M}$). The enhanced binding affinity between phloretin and GLUT3_{exo} results from a fixed outward-facing state of GLUT3_{exo}, whereas WT GLUT3 likely exists at an equilibrium between the outward and inward-facing states, the latter being incapable of binding to phloretin.

5. Line 227, could the authors elaborate on what “chemical property” of maltose is not suitable for optimization?

Maltose, whose chemical structure is shown below, is a disaccharide containing two glucose moieties. It is an abundant metabolic intermediate from starch to glucose.

Any glucose or maltose derivatives will have to compete with endogenous glucose or maltose, which are present at high concentrations, for binding to GLUTs. While glucose may be explored for derivatives with higher affinity, we reasoned that inhibitors using maltose as the scaffold might not be left with many choices for modifications because both the orthosteric and allosteric sites are already occupied by the two sugar rings of maltose. But this statement can be confusing, especially in the absence of experimental validation. We have removed this statement from the revision. We thank the reviewer for identifying this issue.

6. Line 285, what was the membrane cut-off for the concentrator?

The membrane cut-off for the concentrator is 50 KDa. We have included this information in the revised method.

7. Line 296/301, mesophase is not a solution; and the mesophase is not “mixed” with the precipitant solution.

We are sorry for this overlook and we appreciate the kind reminder. The sentence has been corrected to “45 nl mesophase was dispensed on glass plate and then overlaid by 1000 nl crystallization buffer” in the revised manuscript.

8. Line 324, please cite the membrane size for extrusion.

Done.

9. Line 325, please specify the ratio of protein:lipids (molarity or mass, for example).

We have specified the protein:lipids ratio in the original manuscript as “purified protein was incubated with liposomes at a ratio of ~ 1:100 (w/w)”.

We thank this reviewer for their constructive comments.

Reviewer #2:

The major concern from this reviewer is about the prospect of our study on drug discovery. We would like to clarify that:

- 1) Our study serves as a proof-of-concept to demonstrate that our engineered GLUT3exo can be used to validate exofacial GLUT1/3 inhibitors. It is not our scope to report new lead compounds.
- 2) A high-resolution structure of a drug target bound to a lead compound can be invaluable for drug discovery, as it provides the accurate template to perform molecular docking and simulation for lead optimization. SA47-bound GLUT3 represents the FIRST and the ONLY structure of any GLUTs bound to a potent exofacial inhibitor.
- 3) The collaboration was initiated by Sanofi, suggesting the interest of pharmaceutical industry in this work. In fact, our study has contributed to their compound screening.

This reviewer raised several specific points that are addressed below:

1. The authors argue that it will be advantageous to develop exofacial GLUT inhibitors, but, in fact, the developments of the last decade have yielded diverse compound classes with cellular and in vivo activity demonstrating that GLUT inhibitors can be found readily, that appear to penetrate cells and tissues. This may not have been proven explicitly, but then the authors also did not prove that the compounds they identified hit the protein from the outside. For instance, in the GPCR field ant/agonists have been developed that penetrate membranes first and then engage their targets by lateral diffusion.

For medicinal chemistry it is a principal advantage if a biological target is readily accessible on the outer cell surface. For small molecules, this avoids issues with cell penetration and increases the applicability of highly selective alternative modalities like antibodies, nanobodies and peptides, which are dependent on the presentation of their targets on the cell surface.

In addition, it opens new possibilities to use exofacial binding to a receptor to use this interaction for drug depots, half-life extension and drug release.

The idea of using intrinsic recognition systems for drug depots and drug release, exemplified by drugs binding to GLUT1 on erythrocytes as moving depots (Wang et al., 2019) or binders for mannose transporters on macrophages to enable glucose sensing insulin, has been proposed and investigated.

We have expanded the introduction on this potential application in our revised manuscript “*On the other hand, insulin conjugated to GLUT-inhibitors has been employed for drug depots, half-life extension, and drug release, a strategy to mitigate the insulin-induced hypoglycemia in type 1 diabetic mouse model*” (Page 3).

2. The authors claim that their mutant can be used to identify novel inhibitors and refer to a “biased screening of the Sanofi library”. What was the screening set-up, and which readout has been used? Dynamic range, Z'-factor, hit rate and further data are not given. It is unclear whether the mutant can indeed be used for screening, the claim is not substantiated.

We are sorry that this was not made clear enough in the manuscript “A biased screening of the Sanofi library for GLUT1 inhibition delivered hits from several chemical classes as described for example in Lit (WO2017207754, WO2019106122A)”

We virtually screened the Sanofi library by docking the virtual hits and selected compounds. We have changed the manuscript accordingly.

A medium-throughput screening (MTS) was performed for novel motifs interacting with the GLUT1. To this end, around 7000 internal compounds were selected by considering carbohydrate headgroups in analogy to glucose as natural substrate, isolated natural products, a broader carbohydrate screening collection and virtual screening results. This latter virtual screening was carried out using 2D fingerprint similarity employing known active carbohydrates and literature actives for this target. All compounds were tested for GLUT1 inhibition at a 100 μ M concentration, which resulted in a hit-rate of 27.5% (Threshold for confirmed hits: $3 * \sigma = 15.4\%$).

Backscreening for attractive clusters and singletons was carried out, followed by IC_{50} determination for 300 of the most promising analogs, which were selected by initial activity and chemical attractiveness. Dose-response testing resulted in a high hit-rate of 54.6% (164 compounds) with an IC_{50} value < 50 μ M and 6.7% (20 compounds) with an IC_{50} value < 10 μ M. The final round for prioritization was based on profiling data in addition to quality control (i.e., LCMS purity, chemical stability), number of other targets addressed by a compound (frequent hitters) and clustering. A full overview of the distribution of hits from MTS is provided in table below. This then led to the identification of the diazaindazole series as most attractive series for further optimization.

GLUT1 hits from MTS for IC_{50} determination	Tested for IC_{50}	Active with IC_{50} < 50 μ M	Active with IC_{50} < 10 μ M	Percent < 50 μ M	Percent < 10 μ M
HTS-Set	67	38	7	1.1	0.2
Risk-Shared-Collection	29	11	0	0.1	0
VS-SGLT	45	26	0	38.2	0
VS-2016	159	89	12	6.3	0.8

Therefore, the structure of GLUT3-SA47 presented here lays the foundation for structure-guided drug discovery.

3. The authors write that their structure can be used for drug design. This may, in principle, be so, but for GLUT-inhibition, dual GLUT1/3-inhibitors are required (also stressed by the authors). So, how can the current data set be used to devise dual inhibitors? Again, there is a far-reaching claim, but no proof – notably not even for improvement of the identified inhibitors. It would actually be doable to validate for the Bayer compound class which the authors rediscovered because Bayer published a large data set, and the paper contains data for GLUT3. Can structure-design based on the current structure explain these data?

We appreciate the insightful comments. For the first question, our structure of GLUT3-SA47 complex reveals a highly conserved inhibitor binding pocket in both GLUT1 and GLUT3. As shown in the figure below, all residues related to SA47 binding are identical between GLUT1 and GLUT3, whereas Thr28 is replaced by Ile in GLUT2 and GLUT4. Besides, Ser71 is not conserved in GLUT2, with the corresponding locus replaced by an Ala. Supporting the important role of Thr28, single point mutation T28A abolished the binding between GLUT3 and SA47, while S71A mutant led to reduced affinity of SA47. Therefore, our structural model reveals a GLUT1/3-specific druggable pocket that can be exploited for GLUT1/3 dual inhibitor design.

For the second question, our results have provided the molecular basis for the SAR of the reported Bayer compounds (Siebeneicher et al., 2016).

(1) ring A

Introduction of an ortho-methoxy group in **ring A (cmpd 3)** increased the potency against GLUT1, while substitution by either OCF3 (**cmpd 4**) or CF3 (**cmpd 6**) resulted in completely inactive compounds. A methoxy substitution at meta- (**cmpd 8**) or para-position (**cmpd 10**) of **ring A** decreased the GLUT1 potency compared to **cmpd 3**. Adding a fluorine atom to the para-position of the methoxy group (**cmpd 15**) favors GLUT1 association.

These observations can be well explained by our structure. The ortho-methoxy group of **ring A** forms a hydrogen bond with Gln281, which is stabilized by the surrounding H-bond network.

- Introducing ortho-methoxy group to **cmpd 3** can strengthen the interaction between the inhibitor and GLUT1/3, thereby **enhancing** the potency.
- The ortho-methoxy group is thoroughly coordinated by Phe24, Gln280, Gln281, Trp386, Asn409, and Trp410 (within 3.5 Å). Replacing the OCF₃ group in **cmpd 4** may lead to a clash between the fluoride moiety and the side chain of surrounding residues.
- Replacing the CF₃ moiety (**cmpd 6**) not only introduces steric hindrance for ligand binding, but also abolishes the hydrogen bond to Gln281.
- For **cmpd 8** and **cmpd 10**, changing the methoxy substituent to meta- or para-position will lead to a strong steric hindrance to Phe70, Ser71, Trp410, and Asn413.
- The increased affinity of **cmpd 15** is likely caused by an additional H-bond between the added fluorine atom and Thr28 of GLUT3.

Cpd	A	R1	GLUT1 ^b IC ₅₀ [μM]	GLUT2 ^c IC ₅₀ [μM]	GLUT3 ^d IC ₅₀ [μM]
1		F	0.27	1.12	0.68
2		OH	0.007	0.06	0.06
3		OMe	0.025	>10	0.25
4		OCF ₃	>10	>10	>10
5		CN	1.81	5.99	2.05
6		CF ₃	>10	>10	>10
7		F	0.29	1.21	0.54
8		OMe	0.18	1.48	0.32
9		CN	0.039	0.22	0.069
10		OMe	1.77	8.7	1.77
11		F	2.4	>10	10
12		CN	1.2	>10	0.54
13		F	0.01	>10	0.04
14		CN	0.15	1.7	0.17
15		F	0.001	>10	0.01
16		CN	0.02	>10	0.05
17		CONH ₂	0.005	0.10	0.037

(2) **ring B**

Trials that omitted one of the piperazine nitrogen atoms (**cmpd 35 and 36**) resulted in less potent compounds. Substitution adjacent to the ring A attachment point (**cmpd 37**) was tolerated, whereas the corresponding regioisomer (**cmpd 38**) became less potent. Methylene bridges across the piperazine ring (**cmpd 39 and 40**) or switching to piperazine mimic (**cmpd 41**) resulted in less potent compounds. Ring enlargement to 1,4-diazepane (**cmpd 42**) increased the potency.

In the GLUT3-SA47 complex structure, **ring B** is enclosed by Phe24, Gln159, Gln280, Gln281, Asn286, Phe377, Trp386, and Asn413. Except the direct interaction between Gln281 and the nitrogen atom of **ring B**, the distances between **ring B** and surrounding residues (~4-5 Å) are larger than those for **ring A** coordination (within 3.5 Å). Therefore, moderately increasing the volume of **ring B** is tolerable (**cmpd 37, 38 and 42**)

- Changing the conformational state of **ring B** (**cmpd 39, 40 and 41**) might induce steric hindrance.
- Compared to **cmpd 3**, the increased potency of **cmpd 37 and 42** likely resulted from better coordination with the substrate binding pocket.
- Omission of the piperazine nitrogens increases the hydrophobicity of **ring B**, which may cause the energy penalty for ligand accommodation in a hydrophilic environment, accounting for the reduced potency of **cmpd 35 and 36**. In contrast to **cmpd 36**, **cmpd 35** disrupts the H-bond between Gln281 and the nitrogen atom of **ring B**, which may further lower its potency.

Cpd	B	GLUT1 ^b IC ₅₀ [μM]	GLUT2 ^b IC ₅₀ [μM]	GLUT3 ^b IC ₅₀ [μM]
3		0.025	>10	0.25
35		1.9	>10	>10
36		0.47	ND	1.20
37		0.014	1.4	0.12
38		0.12	2.5	0.48
39		2.3	>10	4.1
40		>10	>10	>10
41		4.4	ND	2.2
42		0.004	0.021	0.008

(3) ring C

It was reported that the pyrazole ring of **ring C** could be exchanged for a triazole (**cmpd 46**), imidazole (**cmpd 47**), or imidazole-2-one (**cmpd 48**) without a loss in activity. On the other hand, removing a nitrogen from the 5- or 7-position (**cmpd 43 and 44**) or adding a nitrogen at the 6-position (**cmpd 45**) all lowered the potency.

In our structure, the pyrazole ring of **ring C** is positioned to an open cavity, whereas the other side of **ring C**, the nitrogen atoms at the 5- and 7-positions, are coordinated by Asn315, Glu159, and Gln377 through a network of H-bonds. Modification of the 5-, 6-, or 7-nitrogen may decrease the potency by disrupting these H-bonds (**cmpd 43, 44 and 45**), whereas replacement of pyrazole ring (**cmpd 46, 47 and 48**) has little effect on the activity.

(4) ring D

It was reported that aryl ring connected to the pyrazole (**ring D**) can be substituted with small functional groups in the ortho- (**cmpd 49, 50 and 51**), meta- (**cmpd 52-55**) and para- (**cmpd 56-59**) position without significant loss of GLUT1 inhibition activity. A fluorine atom at the ortho-position (**cmpd 49**) even increased the potency compared to **cmpd 3**. Double substitution at the aryl ring was also feasible, but the electronic behavior and the relative positioning of the substituents was important to keep its activity (**cmpd 60 and 61**). Introduction of a pyridine nitrogen to the benzene **ring D** with (**cmpd 66 and 67**) or without further substitution (**cmpd 63, 64, and 65**) also significantly reduced the potency.

According to our structural observation, **ring D** sits in a large pocket, where it forms a π-π stacking interaction with the side chain of Phe289.

- Adding small functional groups to the ortho- (**cmpd 49, 50 and 51**), meta- (**cmpd 52-55**) or para- (**cmpd 56-59**) position is tolerable, which has little effect to the potency of GLUT1. The increased affinity of **cmpd 49** is likely caused by adding a H-bond between the fluorine substitution and Gln170.
- Double substitution at the aryl ring is tolerable from steric hindrance. Electron-withdrawing group, like fluorine, is preferred as it can maintain the π - π stacking interaction (**cmpd 60 and 61**).
- Introduction of a pyridine nitrogen to the benzene **ring D** will decrease the binding energy of π - π stacking interaction and therefore lower the potency of **cmpd 63-67**.

Cpd	D	R ¹	GLUT1 ^b IC ₅₀ [μM]	GLUT2 ^b IC ₅₀ [μM]	GLUT3 ^b IC ₅₀ [μM]	Cpd	D	GLUT1 ^b IC ₅₀ [μM]	GLUT2 ^b IC ₅₀ [μM]	GLUT3 ^b IC ₅₀ [μM]
3		H	0.025	>10	0.25					
49		F	0.007	1.1	0.04	63		0.51	8.0	0.94
50		OMe	0.05	1.2	0.11					
51		Me	0.01	1.0	0.25					
52		F	0.035	>10	0.12					
53		CN	0.033	nd	0.35	64		0.75	6.0	1.5
54		OMe	0.2	0.9	0.17					
55		Me	0.05	10	0.096					
56		F	0.017	>10	0.046					
57		CN	0.017	>10	0.2	65		1.2	4.4	1.6
58		OMe	0.23	2.6	0.17					
59		Me	1.5	10	3.0					
60		-	0.008	>10	0.09	66		0.08	7.0	ND ^c
61		-	0.03	0.88	0.32	67		0.1	ND	0.34
62		-	0.41	2.1	0.95	68		0.04	2.0	0.12

4. Despite the fact that the authors claim use of their mutant for identification of new inhibitors, the paper does not contain a new class that could be pursued with faith (compounds SA1-5 do not really qualify).

The major scope of our present study is to establish a straightforward tool for exofacial inhibitor validation and to reveal the molecular basis of a representative GLUT1/3 exofacial inhibitor. We chose SA47, rather than the less potent hits that have new chemical scaffold from our screening, for structural investigation as a proof of concept. Our study serves as a framework to aid future identification and validation of new class exofacial inhibitors.

5. There are several GLUT structures published by now and both GLUT1 and -3 have to be considered here (see above; GLUT1-structures with ligands in, are also known, a Bayer paper). This novelty of the structure shown here is limited, and since drug design will need the compound bound to both GLUT1 and -3 (see above), applicability is limited as well. In addition, the strategy to lock transporters into the outward-open conformation by appropriate mutations has been published before by the authors (as they also write), such that this is not novel methodology.

There are four published papers on the structural investigation of GLUT1 and GLUT3 (Deng et al., 2015; Deng et al., 2014; Jiang et al., 2020; Kapoor et al., 2016). Only one of these reported structures of GLUT1 bound to inhibitors (CCB, GLUTi1 or GLUTi2) (Kapoor et al., 2016). Of particular note, all these GLUT1-inhibitor structures are in the inward-facing state, while the molecular basis for GLUT inhibition by exofacial inhibitor remained less well understood. Our structure thus represents the first GLUT structure in complex with an exofacial inhibitor. As discussed in our reply to question 1, the advantages of exofacial inhibitor in drug development require validation tools and a starting structural model. Our structural characterization of GLUT3-SA47, for the first time, elucidated the MOA of an exofacial inhibitor on GLUT3. Given that GLUT1 and GLUT3 share a high sequence similarity (81%) and the SA47 binding pocket is conserved in GLUT1 and GLUT3 (please see question 3, answer 1), our inhibitor-bound GLUT3 structure establishes the framework for virtual screening of exofacial GLUT1/3 dual inhibitors and rational lead optimization. Please be noted that SA47 can inhibit both GLUT1 and GLUT3 with similar affinities in our proteoliposome-based inhibition assay and binding assay (Figure 2b, c).

Before the engineered GLUT3 was generated, there were mainly four ways to distinguish exo- and endo-facial inhibitors (Table below): [1] kinetic simulation of transport assay, [2] inhibition competition assay, [3] structural determination, and [4] virtual docking. The first three approaches are time-consuming and cost-inefficient. The results of virtual docking, on the other hand, must be confirmed experimentally. Our engineered GLUT3 can be stably stored in -80 °C fridge for several months. It provides a fast (less than 1h to finish one measurement) and cost-efficient manner to discriminate exo- and endo-facial inhibitors.

We reported the engineered Xyle variant as a proof-of-concept. Our ensuing study showed that this variant may lead to prominent false negative results owing to its

different substrate preference and varied composition for transport path (Supplementary Figure 1).

On the other hand, XylE, as an *E. coli* protein, is much easier to manipulate. Success with XylE engineering did not warrant a straightforward translation to GLUT1/3. To screen for suitable human GLUT1/3 variants, we had to screen dozens of GLUT1/3 double tryptophan mutants. Eventually GLUT3exo (S64W&I305W) stood out as a stable protein for exofacial inhibitor validation. The study presented in the current report is nothing trivial.

GLUT inhibitors	Class	Strategy
Cytochalasin B (CB) (Deves and Krupka, 1978) (Basketter and Widdas, 1978) (Carruthers and Helgerson, 1991) (Kapoor et al., 2016)	endofacial	[1] zero-trans exit, equilibrium exchange [1] Exit and exchange experiments [1] zero-trans uptake [3] Structure determination
Phloretin (Basketter and Widdas, 1978) (Krupka, 1985a)	exofacial	[1] Exit and exchange experiments [1] zero trans entry, zero trans exit
Maltose (Basketter and Widdas, 1978)	exofacial	[1] Exit and exchange experiments
Androsteneione (Krupka, 1985b)	endofacial	[1] glucose exit experiments
DNTB (May, 1989)	exofacial	[1] zero trans entry, zero trans exit [2] CB/maltose binding study
Androgens, catechins, flutamide (Naftalin et al., 2003)	exofacial	[1] Monitor external glucose affinity (infinite cis K_m) and maximal rate of glucose exit (zero-trans V_m) [4] Molecular modeling
Quercetin (Cunningham et al., 2006)	endofacial	[1] Glucose Efflux [4] Docking studies
GLUT-i1, GLUT-i2 (Kapoor et al., 2016)	endofacial	[3] Structure determination
WZB177 (Ojelabi et al., 2016)	exofacial	[1] zero-trans uptake, zero-trans exit, equilibrium exchange [2] Equilibrium CB binding [4] Molecular docking

The approaches to distinguish the exofacial and endofacial inhibitor are classified into four major categories: [1] kinetic simulation of transport assay, [2] inhibition competition assay, [3] structural determination, and [4] virtual docking.

We thank this reviewer for all the insightful and constructive comments.

Reviewer #3:

This reviewer thinks highly of our study – “*the authors have investigated the molecular basis for inhibiting human GLUTs, in particular they have focused on GLUT3. This is highly relevant study and it should be published. The three-dimensional structure of GLUT3 in complex with a novel inhibitor was determined and the activity was confirmed by uptake measurement and the binding affinity was also determined.*”

They raised a few specific questions that are addressed below:

Major points:

1. *In general, I believe this a solid manuscript, what is lacking is, however, the uptake measurement in wtGLUT3 with the SA47 inhibitor in a comparison with maltose. The authors state that the inhibitor (SA47) binds to GLUT3 in a similar manner as maltose, but since maltose is not suitable for lead optimization, this manuscript is still of clear importance. However, the authors do not show how well the glucose uptake is blocked by SA47 compared to maltose. Thus, the author needs to include maltose as control in Fig 2b.*

In our previous report on the structure of GLUT3-maltose complex, we showed that GLUT1/3 could not be completely blocked by 50 mM maltose (Deng et al., 2015) (Extended data figure 5c). SA47, with its limited water solubility, can only be applied up to 160 μ M. As shown in the table below, SA47 at 160 μ M can inhibit about 85% of the transport activities of GLUT1 and GLUT3. In contrast, GLUT1 and GLUT3 still preserve one third and one quarter of their respective activities in the presence of 50 mM maltose. SA47 is clearly a much more potent inhibitor than maltose for GLUT1/3. We appreciate this insightful comment and have included this discussion in our revised manuscript (Page 11) “*Prior to this study, the only structures of GLUTs with exofacial ligands are GLUT3 bound to maltose in the outward-open and occluded states. Maltose is a physiologically abundant disaccharide that can weakly inhibit glucose transport by GLUT1 and GLUT3. In contrast to the submicromolar range IC₅₀ of SA47, maltose cannot completely inhibit GLUT1 and GLUT3 even when applied at 50 mM*”.

	Transport activity (percentage of Control)	
	GLUT1	GLUT3
SA47 (160 μ M)	~15 %	~13 %
Maltose (50 mM)	~35 %	~25 %

2. *Figure 5 is labelled wrongly, the figure legend does not fit the figure, for instance there is “c” in the legend that is not in the figure.*

We are sorry for this oversight. The manuscript was proofread by all authors before submission. One author rearranged the panels, but forgot to update the legend accordingly. We have corrected the legend of Figure 5.

Minor points:

3. The authors are using mixed code for residues, use one or three letter code consistently in the text and figures.

Three-letter code may be more intelligible for general readers, but may make the figures look crowded. We believe Nature Communications has a defined house style. We will observe the guideline for the format of the text and figures at a later stage.

We thank this reviewer for carefully reading our manuscript and for all their constructive comments.

References:

- Basketter, D.A., and Widdas, W.F. (1978). Asymmetry of the hexose transfer system in human erythrocytes. Comparison of the effects of cytochalasin B, phloretin and maltose as competitive inhibitors. *J Physiol* 278, 389-401.
- Carruthers, A., and Helgerson, A.L. (1991). Inhibitions of sugar transport produced by ligands binding at opposite sides of the membrane. Evidence for simultaneous occupation of the carrier by maltose and cytochalasin B. *Biochemistry-U S A* 30, 3907-3915.
- Cunningham, P., Afzal-Ahmed, I., and Naftalin, R.J. (2006). Docking studies show that D-glucose and quercetin slide through the transporter GLUT1. *J Biol Chem* 281, 5797-5803.
- Deng, D., Sun, P.C., Yan, C.Y., Ke, M., Jiang, X., Xiong, L., Ren, W.L., Hirata, K., Yamamoto, M., Fan, S.L., *et al.* (2015). Molecular basis of ligand recognition and transport by glucose transporters. *Nature* 526, 391-+.
- Deng, D., Xu, C., Sun, P., Wu, J., Yan, C., Hu, M., and Yan, N. (2014). Crystal structure of the human glucose transporter GLUT1. *Nature* 510, 121-125.
- Deves, R., and Krupka, R.M. (1978). Cytochalasin B and the kinetics of inhibition of biological transport: a case of asymmetric binding to the glucose carrier. *Biochim Biophys Acta* 510, 339-348.
- Jiang, X., Wu, J., Ke, M., Zhang, S., Yuan, Y., Lin, J.Y., and Yan, N. (2019). Engineered XyleE as a tool for mechanistic investigation and ligand discovery of the glucose transporters GLUTs. *Cell Discov* 5, 14.
- Jiang, X., Yuan, Y., Huang, J., Zhang, S., Luo, S., Wang, N., Pu, D., Zhao, N., Tang, Q., Hirata, K., *et al.* (2020). Structural Basis for Blocking Sugar Uptake into the Malaria Parasite *Plasmodium falciparum*. *Cell* 183, 258-268 e212.
- Kapoor, K., Finer-Moore, J.S., Pedersen, B.P., Caboni, L., Waight, A., Hillig, R.C., Bringmann, P., Heisler, I., Muller, T., Siebeneicher, H., *et al.* (2016). Mechanism of inhibition of human glucose transporter GLUT1 is conserved between cytochalasin B and phenylalanine amides. *Proc Natl Acad Sci U S A* 113, 4711-4716.
- Krupka, R.M. (1985a). Asymmetrical binding of phloretin to the glucose transport system of human erythrocytes. *J Membr Biol* 83, 71-80.
- Krupka, R.M. (1985b). Reaction of the glucose carrier of erythrocytes with sodium tetrathionate: evidence for inward-facing and outward-facing carrier conformations. *J Membr Biol* 84, 35-43.
- May, J.M. (1989). Inhibition of hexose transport in the human erythrocyte by 5, 5'-dithiobis(2-nitrobenzoic acid): role of an exofacial carrier sulfhydryl group. *J Membr Biol* 108, 227-233.
- Naftalin, R.J., Afzal, I., Cunningham, P., Halai, M., Ross, C., Salleh, N., and Milligan, S.R. (2003). Interactions of androgens, green tea catechins and the antiandrogen flutamide with the external glucose-binding site of the human erythrocyte glucose transporter GLUT1. *Br J Pharmacol* 140, 487-499.
- Ojelabi, O.A., Lloyd, K.P., Simon, A.H., De Zutter, J.K., and Carruthers, A. (2016). WZB117 (2-Fluoro-6-(m-hydroxybenzoyloxy) Phenyl m-Hydroxybenzoate) Inhibits GLUT1-mediated Sugar Transport by Binding Reversibly at the Exofacial Sugar Binding Site. *J Biol Chem* 291, 26762-26772.
- Siebeneicher, H., Bauser, M., Buchmann, B., Heisler, I., Muller, T., Neuhaus, R., Rehwinkel, H., Telsler, J., and Zorn, L. (2016). Identification of novel GLUT inhibitors. *Bioorg Med Chem Lett* 26, 1732-1737.
- Wang, J., Yu, J., Zhang, Y., Kahkoska, A.R., Wang, Z., Fang, J., Whitelegge, J.P., Li, S., Buse, J.B., and Gu, Z. (2019). Glucose transporter inhibitor-conjugated insulin mitigates hypoglycemia. *Proc Natl Acad Sci U S A* 116, 10744-10748.

REVIEWERS' COMMENTS

Reviewer #1 (Remarks to the Author):

The authors have addressed all of my concerns in their revised manuscript.

A minor note:

I applaud the authors' efforts in screening dozens of double-Trp mutants to obtain GLUT3exo as mentioned in the rebuttal letter. I'd encourage the authors to include this major exercise (such as the list of mutants tested) somewhere.

Reviewer #2 (Remarks to the Author):

I have reconsidered the paper under review and taken the authors' comments into consideration.

While some of my questions were answered, the key criticism remains valid. This is a paper with fairly limited impact. The data certainly deserve to be published, but in my opinion not in Nature Communications. The current benchmark in GLUT-inhibition is set by Kadmon: doi.org/10.1016/j.chembiol.2021.10.007. The paper shows that inhibitor development to in vivo active compounds does not need such structures.

Interestingly, Kadmon was acquired by Sanofi. So I am sure, the authors will be well aware of this work.

Reviewer #3 (Remarks to the Author):

I do not have any further comments.